# Interferences of Waxes on Enzymatic Saccharification and Ethanol Production from Lignocellulose Biomass

**DOI:** 10.3390/bioengineering8110171

**Published:** 2021-11-02

**Authors:** Marttin Paulraj Gundupalli, Santi Chuetor, Kraipat Cheenkachorn, Kittipong Rattanaporn, Pau-Loke Show, Yu-Shen Cheng, Malinee Sriariyanun

**Affiliations:** 1Chemical and Process Engineering, The Sirindhorn International Thai-German Graduate School of Engineering, King Mongkut’s University of Technology North Bangkok, Bangkok 10800, Thailand; marttin.g@tggs.kmutnb.ac.th; 2Biorefinery and Process Automation Engineering Center (BPAEC), King Mongkut’s University of Technology North Bangkok, Bangkok 10800, Thailand; santi.c@eng.kmutnb.ac.th (S.C.); kraipat.c@eng.kmutnb.ac.th (K.C.); 3Department of Chemical Engineering, Faculty of Engineering, King Mongkut’s University of Technology North Bangkok, Bangkok 10800, Thailand; 4Department of Biotechnology, Faculty of Agro-Industry, Kasetsart University, Bangkok 10900, Thailand; kittipong.r@ku.th; 5Department of Chemical and Environmental Engineering, Faculty of Engineering, University of Nottingham Malaysia, Semenyih 43500, Selangor, Malaysia; PauLoke.Show@nottingham.edu.my; 6Department of Chemical and Materials Engineering, National Yunlin University of Science and Technology, Yunlin 64002, Taiwan; yscheng@gemail.yuntech.edu.tw

**Keywords:** dewax, biorefinery, lignocellulosic biomass, fermentation, saccharification

## Abstract

Wax is an organic compound found on the surface of lignocellulose biomass to protect plants from physical and biological stresses in nature. With its small mass fraction in biomass, wax has been neglected from inclusion in the design of the biorefinery process. This study investigated the interfering effect of wax in three types of lignocellulosic biomass, including rice straw (RS), Napier grass (NG), and sugarcane bagasse (SB). In this study, although small fractions of wax were extracted from RS, NG, and SB at 0.57%, 0.61%, and 1.69%, respectively, dewaxing causes changes in the plant compositions and their functional groups and promotes dissociations of lignocellulose fibrils. Additionally, dewaxing of biomass samples increased reducing sugar by 1.17-, 1.04-, and 1.35-fold in RS, NG, and SB, respectively. The ethanol yield increased by 1.11-, 1.05-, and 1.23-fold after wax removal from RS, NG, and SB, respectively. The chemical composition profiles of the waxes obtained from RS, NG, and SB showed FAME, alcohol, and alkane as the major groups. According to the conversion rate of the dewaxing process and ethanol fermentation, the wax outputs of RS, NG, and SB are 5.64, 17.00, and 6.00 kg/ton, respectively. The current gasoline price is around USD 0.903 per liter, making ethanol more expensive than gasoline. Therefore, in order to reduce the cost of ethanol in the biorefinery industry, other valuable products (such as wax) should be considered for commercialization. The cost of natural wax ranges from USD 2 to 22 per kilogram, depending on the source of the extracted wax. The wax yields obtained from RS, SB, and NG have the potential to increase profits in the biorefining process and could provide an opportunity for application in a wider range of downstream industries than just biofuels.

## 1. Introduction

Fuels derived from fossil fuels account for about 40% of the world’s total energy consumption [1]. The harmful effects of fossil fuels on the environment have forced society to reduce greenhouse gas (GHG) emissions and use renewable fuels by mixing with fossil fuels [2]. Bioethanol produced from sugars derived from first- or second-generation bioenergy crops (lignocellulosic biomass) can be blended with petrol/gasoline in different ratios (E10 and E85) [3]. Lignocellulosic biomass is the most abundant renewable material, accounting for 64% of the Earth’s terrestrial biomass [4]. In general, lignocellulose is mainly composed of cellulose, hemicellulose, and lignin in the range of 0.35–0.55, 0.20–0.40, and 0.1–0.25 g/g of dry biomass, respectively [5]. It also comprises extracts including wax, pectin, resin, minerals, and ash [6]. Ethanol production from lignocellulosic biomass includes three major processes: pretreatment, saccharification, and fermentation. However, a few studies have reported that including a dewaxing process can increase the saccharification and fermentation efficiency because the mass proportion of wax in biomass is relatively minute and negligible [7,8].

The surface of lignocellulosic biomass is covered with a cuticle layer consisting of wax and its derivatives to form a glossy and white powdery appearance [9]. The wax deposition and its composition vary from plant to plant. Cutin and epidermal wax are composed of hydrocarbons, wax esters (fatty alcohols and acids), aldehydes, ketones, alcohols (primary and secondary), carboxylic acids, and free fatty acids [10]. Intra-cuticular and epicuticular waxes have essential functions in plants, such as (a) protection of the plant from harsh environments (UV and drought), (b) protection from microbes and pathogens, and (c) control of cuticular transpiration [11]. Although plant wax is a minor component in plant biomass, it is used as a precursor for the production of different high-value products in industries—for example, carnauba wax (~2–20 USD/kg), jojoba wax (~22 USD/kg), candelilla wax (~8 USD/kg), and rice bran wax (~18 USD/kg). Wax’s chemical and physical properties, such as low surface tension, high energy content, hardness, adhesive strength, and optical transparency, make it suitable for various uses in industrial applications [12]. In addition, the compositions of epicuticular wax have been associated with chemotaxonomy studies for the identification of plant species [13].

In the present study, the saccharification and fermentation efficiency of different lignocellulosic biomass types (rice straw (RS), sugarcane bagasse (SB), and Napier grass (NG)) with and without wax was studied and compared. These three types of biomass were selected because of their abundancy, global availability, and potential biomass for second-generation energy. Furthermore, after the harvesting seasons in developing countries, these biomass residues are combusted on-field to prepare the field for the next round of cultivation, which becomes a global problem for air pollution, especially PM2.5 and PM10 [14]. Therefore, it is necessary to develop a process for the utilization of these residues to produce value-added products to achieve sustainable economics. In addition, the value-added byproduct (wax) after the dewaxing process was characterized through GC-MS and FTIR to identify potential properties and chemicals as a guideline for product development from value-added wax in the future.

## 2. Materials and Methods

### 2.1. Lignocellulosic Biomass Substrate Collection and Characterization

Lignocellulosic biomass (RS, SB, and NG) was collected from local farms and sugar processing factories located in the central part of Thailand. The collected biomass samples were dried by placing them in a hot air oven at 80 °C until a constant weight was achieved. The particle size of the biomass samples was reduced using a food processor. The particle size uniformity of the biomass samples was achieved by passing the samples through a 20 mesh aluminum sieve with pore size <1 mm. The processed samples were stored in an airtight container until further use. The cellulose, lignin, and hemicellulose contents were determined by following the detergent fiber method [15].

### 2.2. Extraction of Wax

The wax from RS, NG, and SB was extracted using hexane according to the reported protocol [16]. The extraction of wax from the biomass (2% *w*/*v* loading) was carried out in a Soxhlet extractor with continuous reflux for 6 h. The wax and solvent were separated by carrying out evaporation of the organic solvent using a rotatory evaporator with a temperature maintained at 45 °C ± 2.0. The wax content and dewaxed solids were determined using Equations (1) and (2).
(1)$$D,\mathrm{mg}=\frac{W_{1}-W_{2}}{W_{2}}\;\times \;100$$
(2)$$Wax,\;\mathrm{mg}=\frac{W_{3}}{W_{1}}\;\times \;100$$
where *D* is the percentage of residual solid after dewaxing, *W*_1_ is the weight of the biomass (mg) collected after Soxhlet extraction, *W*_2_ is the initial weight of the biomass (mg), and *W*_3_ is the weight of the wax (mg) collected after evaporating the hexane solvent using a rotary evaporator.

### 2.3. Enzymatic Saccharification

Saccharification of control (C) or untreated samples and dewaxed samples (D) was performed in a 15-milliliter centrifuge tube. A saccharification mixture containing 4 mL of 50 mM citrate buffer (pH 4.8), biomass loading (2.5% *w*/*v*), and 40 μL of 2 M sodium azide (2% *w*/*v*) (Sigma-Aldrich, MO, USA) was made [17]. The saccharification of the C and D samples was carried out using β-glucosidase (Megazyme, MI, USA) and Celluclast 1.5L^®^ (Sigma-Aldrich, MO, USA). As reported in previous studies, 100 μL of β-glucosidase and 350 μL of Celluclast 1.5L^®^ (Sigma-Aldrich, MO, USA) per gram of dried sample were added to the tube [18]. The saccharification process was performed at 50 °C for 72 h while maintaining tube rotation at 150 rpm. The sugars released during the 72-h period were quantified using a modified 3,5-dinitrosalicylic acid (DNS) assay method [19,20]. The process was repeated thrice without the addition of sodium azide for the preparation of hydrolysate for ethanol production.

### 2.4. Production of Ethanol

Fermentation of the hydrolysate collected after saccharification was carried out in an Erlenmeyer flask. A yeast culture (*Saccharomyces cerevisiae* TISTR 5606) was obtained from the Thailand Institute of Science and Technology Research (TISTR), Thailand [18]. The yeast culture was grown in a 100-milliliter Erlenmeyer flask containing yeast–peptone–dextrose (YPD) broth (HiMedia Laboratories, Mumbai, India). Yeast growth was monitored with a UV–visible spectrophotometer (PG Instruments Limited, Leicestershire, UK) at 600 nm until the absorbance was 1.0 ± 0.2 (yeast growth phase at log phase). To a 50-milliliter flask, 19 mL of the hydrolysate and 1 mL of the cultured yeast were added. In addition, 5% *w*/*v* sucrose and 1% *w*/*v* yeast extract were added to the flask. The medium was incubated at 32 °C for 48 h in a temperature-controlled orbital shaker with rotation maintained at 150 rpm. At the end of the fermentation process, the medium was centrifuged for 10 min at 8000 rpm. The concentrations of the ethanol produced after 48 h were determined using GC-MS [21].

### 2.5. Scanning Electron Microscopy

The changes in the surface morphology of C and D samples for different biomass types were scanned and photographed using a scanning electron microscope (SEM) (Jeol, JSM-5410LV, Tokyo, Japan). The biomass samples were attached to aluminum stubs using double-sided carbon tape. The C and D samples of RS, SB, and NG were sputter-coated with gold alloy. The scanned images of C and D samples were then studied and compared.

### 2.6. Fourier-Transform Infrared Spectroscopy

Fourier-transform infrared spectroscopy (FTIR) of the biomass and wax samples was carried out to determine the changes in functional groups using an FTIR spectrum instrument (Spectrum, Perkin Elmer, MA, USA). The FTIR analysis of samples was carried out by preparing biomass KBr discs (pellets) containing 200 mg solids and 2 mg KBr. Each disc was placed inside the instrument and scanned and recorded 16 times at a wavelength range of 4000–400 cm^−1^ with a resolution of 4 cm^−1^.

### 2.7. Wax Composition Analysis by GC-MS

A compositional analysis of the wax extracted from RS, NG, and SB was performed using a gas chromatograph mass spectrometer (GC-MS). Before the analysis, 10 mg of wax was added in 20 mL of methanol with 1.0 M of NaOH. The tube was placed in a water bath for 1 h, with the temperature maintained at 70 °C for the transesterification process. The tube was allowed to cool down and centrifuged at 10,000 rpm for 30 s. The clear supernatant enriched with esterified products was mixed with hexane and vortexed for product transfer. The top layer with hexane and esterified products was collected and transferred to another tube. The hexane was evaporated to determine the weight of the esterified products. The dried sample was resolubilized with hexane to determine the composition of the wax using GC-MS. The sample was analyzed in a DB-Wax column (30 m × 0.25 mm, 0.25 µm) with helium as the carrier gas. The injector and detector temperatures were maintained at 220 and 200 °C, respectively. The temperature program of the GC-MS oven was set as follows: 100 °C, 10 °C/min to 200 °C, 15 °C/min to 220 °C, and hold at 220 °C for 5 min. The injection volume and injection mode were 1 µL and split (split ratio of 1:30), respectively.

### 2.8. Ethanol Determination Using GC-MS

The ethanol concentration was determined using a GC-FID (GC-2010, Shimadzu, Kyoto, Japan) equipped with an FID detector and a DB-Wax column (30 m × 0.25 mm, 0.25 µm). The conditions for the column oven were as follows: (i) 40 °C for 4 min, (ii) 100 °C at 5 °C/min, and (iii) 200 °C at 10 °C/min [22]. The ethanol concentration in the supernatant was determined and calculated from the calibration curve generated from different concentrations of ethanol (0.1–1% *v*/*v*) prepared from 99.999% ethanol solvent. 

### 2.9. Statistical Analysis

The data obtained before and after dewaxing were analyzed for statistical significance using Origin Pro (OriginLab, MA, USA). The differences in significance between control (C) and dewaxed (D) samples were evaluated by *t*-test (paired), with a *p*-value lower than 0.05 considered significant.

## 3. Results and Discussion

### 3.1. Wax Yield and Lignocellulosic Biomass Composition

The wax yields obtained from Soxhlet extractions with hexane of RS, NG, and SB are listed in Table 1. Wax yields of 0.57%, 0.61%, and 1.69% were determined for RS, NG, and SB, respectively. Compared to the whole biomass weight fraction, the wax compositions of these three types of biomass were small mass fractions, and the weights of biomass before and after dewaxing did not change much (Figure 1). In a similar study, wax was extracted via hexane reflux action from the leaves of different banana plant species (*Musa* sp.). Wax yields of 0.58%, 1.05%, and 1.41% were reported for the leaves of *M. liukiuensis, M. acuminate*, and *M. chiliocarpa*, respectively [23]. Other studies reported wax yields of 0.41% from wheat leaves [24] and 0.2% from flax straw [25] after using n-hexane as the extraction solvent.

Lignocellulosic biomass characterizations were performed to determine the cellulose, lignin, and hemicellulose contents in RS, NG, and SB (Figure 2). The characterizations of the plant components showed slight changes in the contents of cellulose and hemicellulose for dewaxed samples compared to control samples. Reductions in cellulose content of 0.10%, 2.01%, and 0.06% caused by dewaxing in RS, NG, and SB, respectively, were observed, as well as decreases in hemicellulose content of 2.02%, 0.40%, and 3.42%, respectively. Similarly, the lignin content in dewaxed samples of NG and SB was also decreased by 0.35% and 0.24%, respectively. On the other hand, the lignin content in RS increased by 4.71% after dewaxing or by 1.79-fold compared to the control sample. These changes in mass fractions suggest that the removal of wax slightly affected the contents of cellulose, hemicellulose, and lignin in RS, NG, and SB. The statistical significance study showed that the changes in the contents of cellulose and hemicellulose for dewaxed RS, NG, and SB samples were not significant (*p* > 0.05). Similar significance was evaluated for lignin of dewaxed NG and SB. However, it was noted that the change in the lignin content for the RS dewaxed solid was significant compared to the control (*p* < 0.05). This observation agreed with another study performed by Jyoti et al., which reported a change in cellulose, hemicellulose, and lignin contents in raw jute and dewaxed jute. The cellulose, hemicellulose, and lignin contents reduced by 1.059-, 1.116-, and 1.040-fold, respectively, for dewaxed jute [30]. Previously, the solubility of lignin in organic solvents was found to depend on the lignin type and the polarity of the solvent; for example, kraft lignin has better solubilization in a protic organic solvent than in an aprotic solvent [31]. Therefore, the difference in the lignin composition of RS after dewaxing by hexane suggests the variation in the chemical compositions of wax and lignin among the three types of biomass tested in this work.

### 3.2. SEM Image Analysis

To understand the effect of dewaxing on the morphology of RS, NG, and SB, an SEM image analysis was performed by comparing C and D samples (Figure 3). Wax layers are commonly found on the outer surface of the sheath, stem, grain, and leaf and appear as a white powdery and shiny appearance on the plant surface [32]. Removal of the wax can change the hydrophobicity and water accessibility of the biomass surface. It was noted that the surfaces of the RS and NG (C samples), as shown in Figure 3a,e, were rough with dense arrangements, while the surface of the SB (C sample) was thick and dense (Figure 3c). The outer surface of the C sample was smooth, which can be attributed to the wax layer present on the surface. The wax present on the surface prevents loss of water through evaporation and protects the plant from pathogens [33]. A change in the surface morphology was observed after wax was removed from the surfaces of the RS, NG, and SB (D samples). The SEM image analysis for D samples of RS, NG, and SB is shown in Figure 3b,d,f, respectively. It was noted that the surfaces of the RS and NG (D samples) were rougher and thinner due to wax removal.

### 3.3. FTIR Analysis

The functional groups of the extracted waxes obtained from RS, NG, and SB were analyzed by using FTIR to differentiate their characteristics (Figure 4a). Based on the FTIR spectra, it was observed that there were many peaks representing the functional groups of chemical compositions in the plant waxes. The peaks at 3428 cm^−1^ in the wax samples could be attributed to the stretching vibrations caused by the OH group in free fatty acid (carboxyl group) and alcohols. This peak appeared as the major peak in the SB wax but had a lower intensity in the RS and NG waxes, suggesting the abundance of fatty acid and alcohol derivatives in SB. The peak at 2923 cm^−1^, corresponding to the presence of alkyl alcohol, esters, and alkyl acids, was observed as a major peak in the wax samples of RS, NG, and SB [34]. Another major peak at 2850 cm^−1^ was attributed to the strong stretching of the CH2 (symmetric) bonds found in the hydrocarbons [35] and aliphatic compounds (alkanes) in waxes [29]. The peak at 1710 cm^−1^, which corresponds to the strong stretching of C=O bonds in acid or aldehyde carbonyl compounds (conjugated) [36], and that at 1460 cm^−1^, which corresponds to C–H bending or scissoring long-chain alkanes [37], were observed with high intensities in the RS and SB waxes but with low intensity in NG. Likewise, the transmittance band observed for the RS and SB waxes at 729 cm^−1^ was attributed to the CH2 bending vibration [38]. Altogether, these profiles of transmitting peaks suggest the variations in the chemical compositions of RS, NG, and SB waxes.

To observe the changes in functional groups of biomass before and after dewaxing, the C and D samples of RS, NG, and SB were analyzed by FTIR analysis (Figure 4b–d). It is presumed that the different spectral profiles between C and D samples could be due to the functional groups of the waxes when the dewaxing process has no effect on biomass compositions. Reductions in the intensity of the major transmittance peak at 3428 cm^−1^, attributed to the strong stretching vibration caused by the OH bond in the biomass, were observed in D samples of SB and RS compared to their corresponding C samples. The vibration is caused due to the intermolecular and intramolecular hydrogen bonding present in the cellulose structure and a phenolic group of lignin [39]. Similarly, the peak intensities at 2923 and 2850 cm^−1^ were reduced in D samples of RS, NG, and SB compared to their C samples. The transmittance peak at 2923 cm^−1^ corresponds to the medium stretching vibration caused by the CH and CH2 bonds in the aliphatic and aromatic compounds present in lignin and cellulose polymers [40]. The peak at 2850 cm^−1^ represents the ester linkages of hemicellulose and lignin. Furthermore, the peak at 1250 cm^−1^, corresponding to the strong stretching bond exhibited by the C-O bonds present in the aromatic layer (lignin) of plants, was observed in both C and D samples of the three types of biomass [41]. Interestingly, the intensities of two peaks at 1055 and 1640 cm^−1^ were reduced in D samples of SB compared to C samples, but the intensities of C and D samples of RS and NG were similar. The transmittance peak at 1055 cm^−1^ corresponds to the strong stretching vibrations due to C-O and C-C bonds in cellulose and hemicellulose structures [42], and the peak at 1640 cm^−1^ represents the stretching of the aromatic benzene ring in lignin [43]. These spectral peaks observed in the C and D samples of the three types of biomass suggest the different influences of the dewaxing process on each type of biomass.

Next, the FTIR spectra of biomass and wax were compared to evaluate the effect of dewaxing on biomass. Among the above-listed transmittance peaks, the peaks at 2850 and 2923 cm^−1^ were observed in both biomass and wax, and their intensities were reduced in D samples of RS, NG, and SB, implying that this reduction was due to the removal of wax components from the biomass. It should be noted that three peaks, at 3428, 1640, and 1055 cm^−1^, were found in biomass samples but not in wax, and their intensities were reduced only in the D sample of SB compared to the C sample. Altogether, the comparisons of the FTIR spectra of biomass and wax suggest that the dewaxing process not only removes wax components from lignocellulosic biomass but also modifies the functional groups in the cellulose, hemicellulose, and lignin of SG biomass.

### 3.4. Enzymatic Saccharification

Wax interference studies on enzyme accessibility to cellulosic structures were conducted during the saccharification of C and D samples through measurement of reducing sugars. The saccharification of the C and D samples was performed for 72 h, as shown in Figure 5. Different yields of sugar were observed for C and D samples of RS, NG, and SB. A higher sugar yield was observed for D samples compared to C samples in the case of all three biomasses. The sugar yield was increased by 1.17-, 1.05-, and 1.35-fold for the RS (D), NG (D), and SB (D) samples, respectively. It was noted that the change in the sugar yield was not significant for RS and NG (*p* > 0.05), whereas the change in the sugar yield for SB was significant (*p* ≤ 0.05). Qi and colleagues (2016) reported a similar observation with an increase in sugar by 1.37-fold during the saccharification of C and D samples of SB [29]. In this study, the best improvement in sugar yields occurred in SB. This saccharification result could be related to the FTIR analysis in the previous section showing that dewaxing not only removes wax from biomass but also modifies the SB biomass. Additionally, the cellulase accessibility to cellulose was studied and reported by performing adsorption studies. The adsorption of cellulase enzyme on an untreated oil palm empty fruit bunch was lower compared to dewaxed biomass. This was assumed to have been caused by the hydrophobic interaction between cellulose and wax [44]. Therefore, it is suggested from the results that the removal of wax can increase saccharification efficiency.

### 3.5. Ethanol Production

Fermentation of the C and D samples was performed to understand the significance of wax removal, as shown in Figure 6. Ethanol fermentation of the hydrolysates of C and D samples was performed for 48 h. The ethanol production of C and D samples was reported after subtracting the ethanol production from sucrose. The ethanol yields obtained from RS, NG, and SB were higher in the case of D samples compared to C samples. The maximum ethanol yield was observed for D samples belonging to NG, followed by RS and SB. The ethanol yield increased by 1.11-, 1.05-, and 1.23-fold after wax removal from RS, NG, and SB, respectively. This improvement in ethanol production has a similar trend to that in enzymatic saccharification in that the dewaxing process had the highest effect on SB. The statistical significance study showed that the ethanol yields from dewaxed samples of RS, NG, and SB were significant since the *p*-value was less than 0.05. However, the change in the ethanol yield for the dewaxed SB sample was highly significant since the *p*-value was less than 0.0005. The results of this experiment show that wax in plant materials could interfere with fermentation efficiency. In another study, the ethanol yield from sorghum grains increased by 1.23-fold after dewaxing, resulting in a 23.3% improvement [45]. Similarly, a 21% ethanol yield from dewaxed wheat straw was reported [46]. The ethanol yield was lower for C samples as lower concentrations of sugars were released into the hydrolysate during saccharification. To date, there are very few studies related to the effect of dewaxing on ethanol production from untreated lignocellulosic biomass because the mass fraction of wax is considered to be a minor component.

### 3.6. Wax Compositional Analysis by GC-MS

The composition of waxes extracted from RS, NG, and SB was analyzed for organic compounds using GC-MS. These wax compositions were classified into fatty acid methyl esters (FAMEs), fatty acids (FAs), alcohols, alkanes, ketones, and other trace compounds (Figure 7). The compositional analysis of waxes from RS, NG, and SB showed that the organic compounds differ depending on the biomass. FAMEs were identified to be the major components in the extracted waxes from all types of biomass, constituting 64.67%, 84.45%, and 88.65% of the RS, NG, and SB waxes, respectively. The alkane compound amount in the RS-extracted wax was higher (16.85%) compared to SB (10.23%) and NG (3.04%). Furthermore, the yield of alcohol-related organic compounds was high in NG (9.6%) compared to RS (6.4%) and SB (1.11%) wax. The FA proportion of RS was 10.10% higher than that of NG (2.13%). The wax compositions were compared with those reported in several studies and are summarized in Table 2.

Intra-cuticular and epicuticular wax is composed of a mixture of long carbon chain molecules, which includes esters, fatty acids, alcohols, alkanes, aldehydes, and ketones, with carbon atoms varying from C_22_ to C_36_ [49]. In the present study, the carbon atoms varied for RS, NG, and SB as C_13_–C_41_, C_10_–C_34_, and C_11_–C_44_, respectively. As mentioned, the profiles of wax compositions, in qualitative and quantitative terms, are considered as a unique fingerprint of plant species [13]. Here, FAME derivatives were identified to be the major components of RS, NG, and SB wax. The carbon atoms of FAMEs for RS, NG, and SB varied between C_13_–C_24_, C_10_–C_24_, and C_13_–C_24_, respectively. The major FAME compounds in RS were hexadecanoic acid methyl ester (C_17_) (39.8%wt), 9-octadecenoic acid methyl ester (C_19_) (6.0%wt), methyl,18-methylnonadecanoate (C_21_) (6.8%wt), docosanoic acid methyl ester (C_23_) (5.4%wt), and tetracosanoic acid methyl ester (C_24_) (4.6%wt). The dominant FAME compounds in wax extracted from NG were hexadecanoic acid methyl ester (C_17_) (21.2%wt), 9,12-octadecadienoic acid methyl ester (C_19_) (21.7%wt), and 9,12,15-octadecatrienoic acid methyl ester (21.6%wt). Meanwhile, hexadecanoic acid methyl ester (C_17_) (47.5%wt), 9,12-octadecadienoic acid methyl ester (C_19_) (11.1%wt), and nonadecanoic acid, 18-methyl, methyl ester (C_21_) (6.1%wt) were dominant in SB. From these most abundant FAMEs in SB, RS, and NG, it could be observed that both RS and SB waxes were composed of long-chain saturated FAMEs, whereas the NG wax was made of long-chain unsaturated FAMEs. This chemical profile analysis correlates well with the FTIR results of wax samples (Figure 4a) showing that the FTIR spectral peaks of SB and RS were more similar compared to NG.

Among the alcohols, phytol (C_20_) showed a higher yield in wax extracted from RS (2.2%wt) and NG (9.0%wt). Phytol, classified as a terpenoid, is used as a precursor for the production of vitamins E and K_1_ in synthetic forms [50]. It is also a precursor for the manufacture of fragrances, shampoos, detergents, soaps, cleaners, and cosmetics [51]. On the other hand, tetratriacontane (C_34_) was among the alkane compounds with a high yield in RS (9.47%wt) and SB (6.27%wt), suggesting that the wax chemical compositions of these two plants are dominated by saturated hydrocarbon compounds. Tetratriacontane (C_34_) is also a long-chain alkane compound with application in biorefinery industries. It was reported previously that long-chain alkane compounds in wax, such as C_22_, C_27_, C_29_, and C_31_ alkanes, function to protect water loss of fruits to prolong the shelf life [49]. FAMEs, FAs, and alkanes were the major groups of compounds identified in the waxes extracted from RS, NG, and SB. The results of the wax composition analysis correlate with the FTIR spectra shown in Figure 4. Again, the correlation of the FTIR spectra and the GC-MS profiling analysis of wax compositions showed that the waxes of RS and SB have closer relationships in terms of chemical composition and abundancy than that of NG.

Though the wax yield is low in lignocellulosic biomass, wax has several compounds that have potential applications in commercial and biorefinery industries. A material balance of the ethanol and wax yields from the present study is represented as a process flow diagram in Figure 8 to understand the significance of the present study. Based on the conversion rate of the dewaxing process and fermentation, the yields of wax from RS, NG, and SB were 5.64, 17.00, and 6.00 kg/ton, respectively. Furthermore, the ethanol yields from RS, NG, and SB were 4.91, 5.24, and 15 kg/ton, respectively. The NREL estimated the cost of ethanol to be around 1.85 USD/liter by combining the price of biomass feedstock, saccharification enzymes, and capital cost [50]. The current gasoline price is around 0.903 USD/liter, making ethanol more expensive than gasoline. Therefore, to reduce the cost of ethanol in biorefinery industries, the commercialization of other valuable products such as wax should be considered. As mentioned, the cost of natural wax ranges from 2 to 22 USD/kg, depending on the source of the extracted wax. The amount of wax produced from RS, SB, and NG can contribute to the revenue in the production process of ethanol. Therefore, this study’s experimental results could provide useful information for biorefineries’ products and process design to improve the profit from lignocellulosic biomass that can contribute to various industries, not just focusing on biofuels.

## 4. Conclusions

The effect of dewaxing lignocellulosic biomass (RS, NG, and SB) on the efficiency of saccharification and fermentation was studied. Enzymatic saccharification of dewaxed RS, NG, and SB samples showed an increase in reducing sugar by 1.17-, 1.04-, and 1.35-fold, respectively, compared to C samples. Similarly, wax removal increased the ethanol yield during saccharification by 1.11-, 1.05-, and 1.23-fold in RS, NG, and SB, respectively. Both saccharification and ethanol fermentation experiments without further biomass pretreatment suggested the interference effect of wax, although it is only a trace mass fraction in biomass. The FTIR analysis also provided evidence of modifications in the biomass composition caused by dewaxing, especially in functional groups’ levels, related to dissociations of lignocellulose fibrils. The chemical composition analysis of waxes by GC-MS showed that the concentration of hexadecanoic acid methyl ester (FAME) was highest in the wax extracted from SB, followed by RS and NG. The profiles of FAMEs and alkane derivatives suggested the close similarity between RS and SB, being made of saturated hydrocarbon compounds. These wax compounds have different applications in various commercial and biorefinery industries. The amounts of wax obtained from RS, SB, and NG and their values in the current markets have the potential to improve the revenue of ethanol produced from lignocellulosic biomass.

## Figures and Tables

**Figure 1 bioengineering-08-00171-f001:**
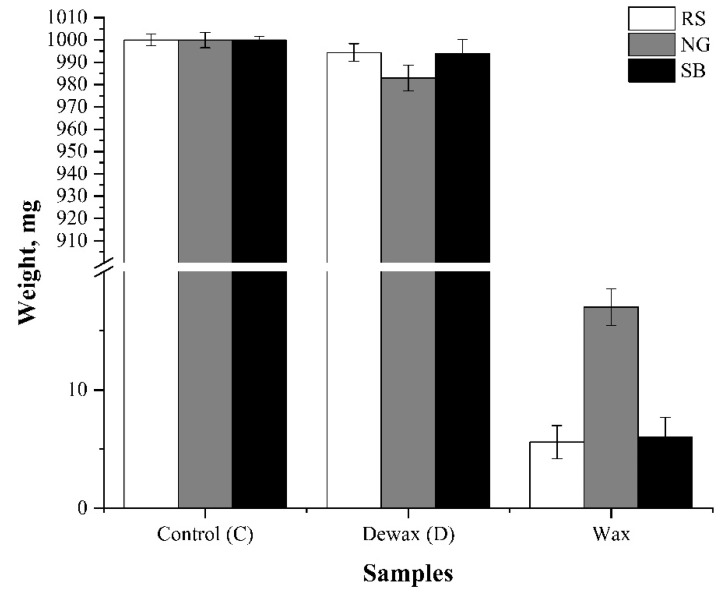
Wax yields of three types of biomass. RS—rice straw; NG—Napier grass; SB—sugarcane bagasse.

**Figure 2 bioengineering-08-00171-f002:**
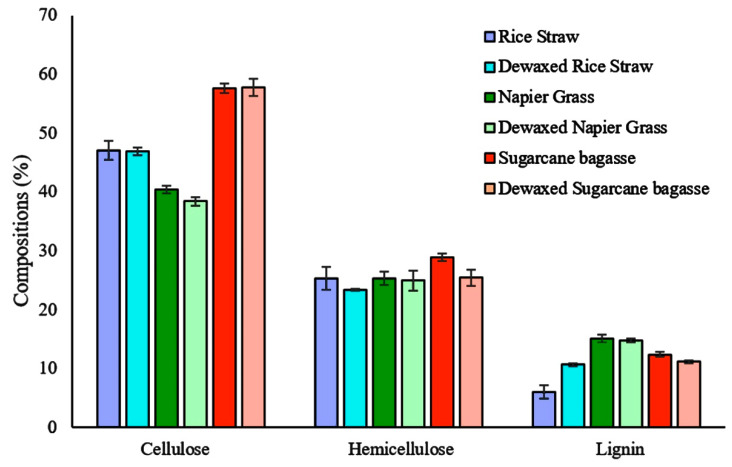
Compositional analysis of different lignocellulosic biomass types before and after the dewaxing process.

**Figure 3 bioengineering-08-00171-f003:**
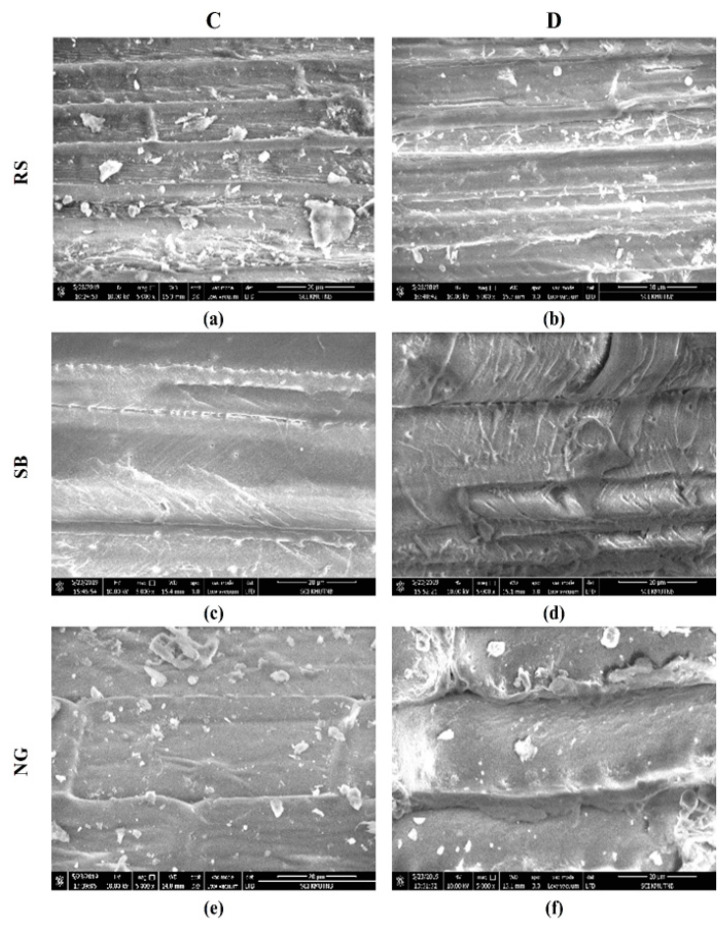
SEM image analysis of control (C) and dewaxed (D) samples of rice straw (**a**,**b**), sugarcane bagasse (**c**,**d**) and nepier grass (**e**,**f**).

**Figure 4 bioengineering-08-00171-f004:**
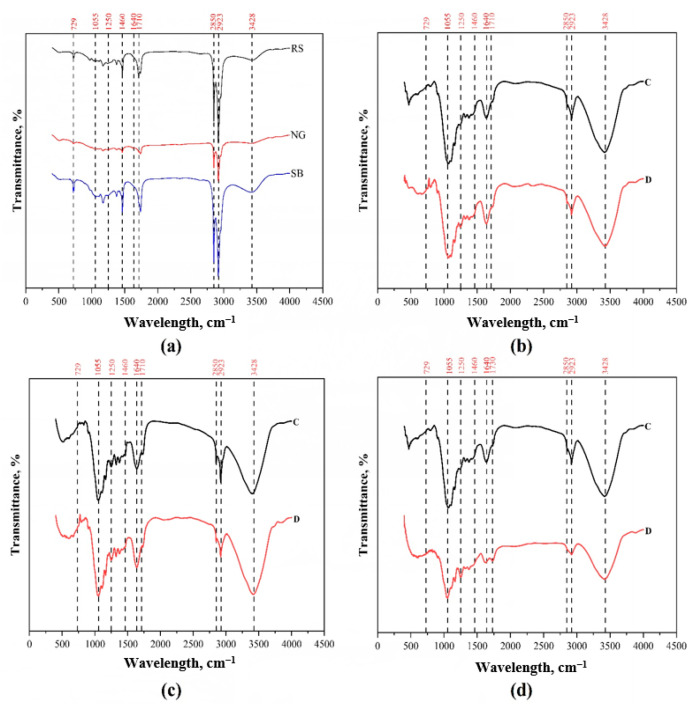
FTIR spectra for wax and different lignocellulosic biomass types. (**a**) IR spectra of wax extracted from RS, NG, and SB; (**b**) rice straw (RS); (**c**) Napier grass (NG); (**d**) sugarcane bagasse (SB).

**Figure 5 bioengineering-08-00171-f005:**
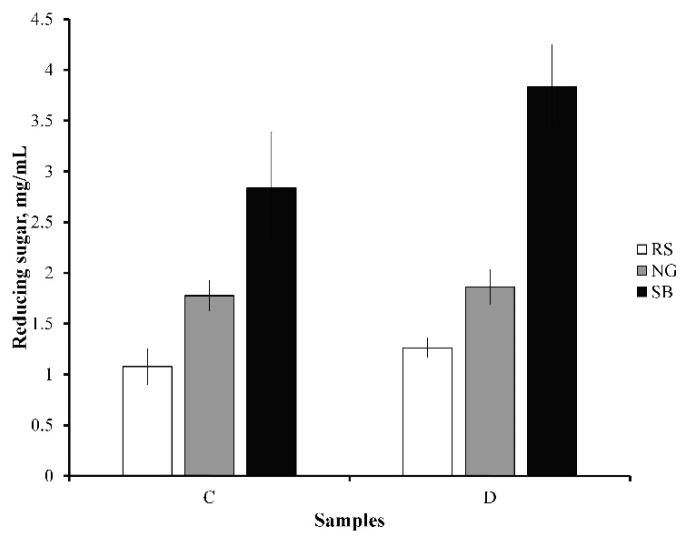
Enzymatic saccharification of control (C) and dewaxed (D) samples. RS—rice straw; NG—Napier grass; SB—sugarcane bagasse.

**Figure 6 bioengineering-08-00171-f006:**
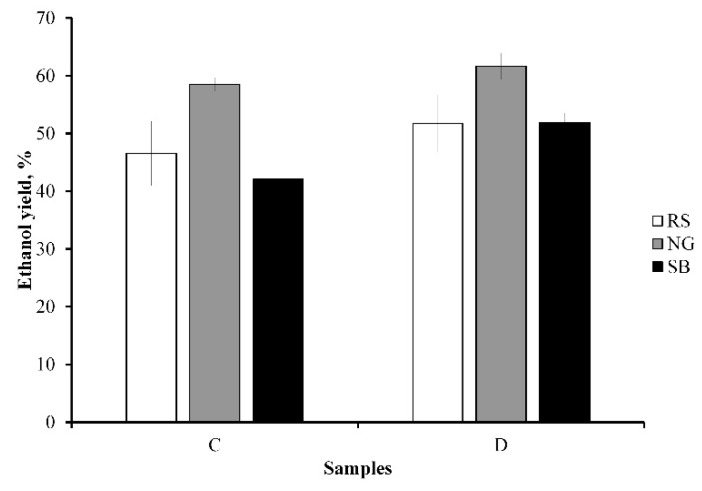
Ethanol production from control (C) and dewaxed (D) samples of rice straw (RS), Napier grass (NG), and sugarcane bagasse (SB).

**Figure 7 bioengineering-08-00171-f007:**
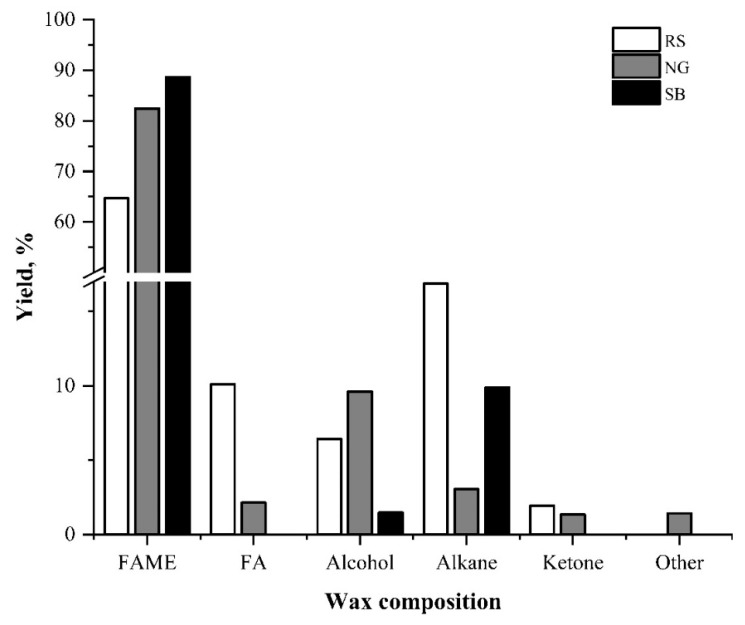
Chemical groups identified by GC-MS analysis of wax extracted from different biomass types using hexane. FAME—fatty acid methyl ester; FA—fatty acid; RS—rice straw; NG—Napier grass; SB—sugarcane bagasse.

**Figure 8 bioengineering-08-00171-f008:**
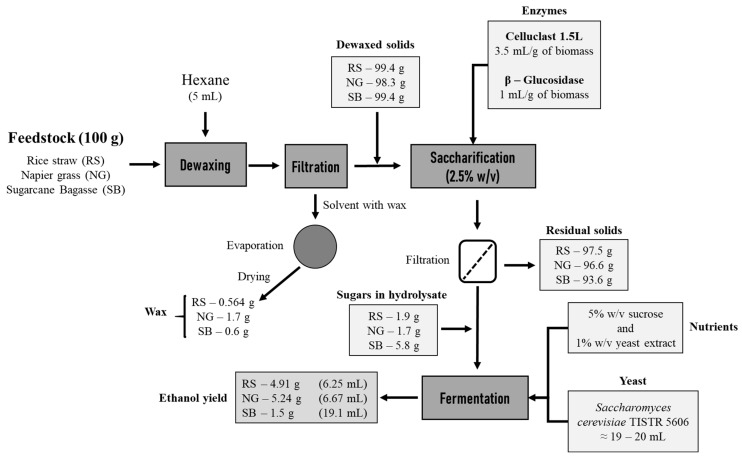
Material balance for ethanol and wax yields from dewaxed solids.

**Table 1 bioengineering-08-00171-t001:** Wax from different lignocellulosic biomass types extracted using n-hexane solvent.

Lignocellulosic Biomass	Wax Yield, %	References
*M. liukiuensis*	0.58	[23]
*M. acuminate*	1.05	
*M. chiliocarpa*	1.41	
Wheat leaves	0.41	[24]
Flax straw	0.20	[25]
Flax processing waste	4.00	[26]
Apple peel pomace	3.98	[27]
*Tamarix nilotica* leaves	0.70	[28]
Sugarcane bagasse	1.20	[29]
Rice straw (RS)	0.57 ± 0.080	Present study
Napier grass (NG)	0.61 ± 0.035	
Sugarcane bagasse (SB)	1.69 ± 0.092	

**Table 2 bioengineering-08-00171-t002:** Studies related to compositional analysis of wax extracted from different lignocellulosic biomass types.

Lignocellulosic Biomass	FAME, %	FA, %	Alcohol, %	Alkane, %	Ketone, %	Other, %	References
Carnauba	54	na	12	1	na	23	[47]
Sugarcane bagasse	66.26	4.58	0.22	28.83	na	0.11	[48]
Wheat straw	11	25	20	na	na	na	[49]
*Pyrus pyrifolia*	3.45	7.09	40.72	24.47	na	43.67	[50]
Blueberry	na	2.8	3.2	1.3	16.4	7.8	[51]
*Citrus sinensis*	na	7.8	0.16	11.5	na	na	[52]
Rice straw (RS)	64.68	12.03	6.43	16.85	1.93	0.00	Present study
Napier grass (NG)	82.46	3.48	9.60	3.05	1.34	1.41	
Sugarcane bagasse (SB)	88.65	nd	1.47	9.88	nd	nd	

na—not available; nd—not detected.

## Data Availability

Not applicable.

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
