# Peer review of "Interferences of Waxes on Enzymatic Saccharification and Ethanol Production from Lignocellulose Biomass"

_bioengineering, 2021, doi:10.3390/bioengineering8110171_

Round 1

Reviewer 1 Report

The manuscript entitled "Interferences of Waxes on Enzymatic Saccharification and Ethanol Production from Lignocellulose Biomass” is about a dewaxing process carried out to three types of lignocellulosic biomass and its effect on saccharification and ethanol production. The importance of the work resides in the need for prepare cellulosic material for enzymatic treatment to yield reducing sugars for the synthesis of ethanol by yeast Saccharomyces cerevisiae TISTR 5606.

This study presents results about biomass composition: cellulose, hemicellulose and lignin content; wax extraction and its characterization; IR analysis of both, biomass and wax; biomass saccharification and yeast fermentation to produce ethanol. The flow process diagram is an important analysis for material balance but it can be improved for better understanding of the overall process.

Some specific comments regarding the research report is shown in attached filed.

Author Response

Thank you for your kind suggestion to improve our manuscript and provide opportunity to revise for us. Please see our response in the attached file. 

Reviewer 2 Report

This original article deals with the study of the effect of waxes within the enzymatic hydrolysis of lignocellulosic materials for the production of bioethanol. The article is in accordance with the journal’s scope, however, some points must be improved before its publication:

  • Lines 107-109. There are two sentences describing the same, please, remove one of them.
  • Section 2.4. Why adding sucrose to the already sugar-rich hydrolysate? Was the amount of ethanol produced from sucrose subtracted from the final ethanol production?
  • 1/ table 1. Was the extraction performed in one replica? I see no standard deviation.
  • Lines 185-190. The cellulose, hemicelluloses and lignin content varies regarding the initial content? Fig. 2 displays composition in %wt, which I suppose is measured as g/100 g of initial biomass for non-dewaxed biomass and g/100 g of dewaxed biomass? Please, put in the same unit. Additionally, a statistical analysis (one-way ANOVA followed by Tukey’s test) could facilitate the explanation of that graph.
  • Line 208. There is a reference which is not in the format of the journal.
  • Statistical analysis should be added to Fig. 5 and Fig. 6, comparing both control and dewaxed samples.
  • I would suggest to enhance the novelty of this work since it is not well reflected in the manuscript.

Author Response

(The authors gave the same response as above.)

Reviewer 3 Report

This original article discusses the effect of waxes on the enzymatic hydrolysis impedance for bioethanol production. Although it comprises the scope of the journal, several points have to be accomplished to publish the manuscript: - Line 107 and following discuss the same. Modify. - Data does not reflect standard deviation, please add. I think a statistical analysis may improve the quality of the manuscript. - The novelty of the manuscript is lacking. I suggest to reform the abstract to give a better overview of the work. - Check the references, I think some are not in the journal’s format.

Author Response

(The authors gave the same response as above.)

Round 2

Reviewer 1 Report

The manuscript has been improved. This revised version is presented with better quality in both, the text and in the content; the results, data presentation and its discussion are clearer.

Details of ethanol determination and flow precess diagram are clearly presented.

Reviewer 2 Report

It can be accepted in the present form.

Reviewer 3 Report

Accepted